# LayerDAG: A Layerwise Autoregressive Diffusion Model for Directed Acyclic Graph Generation

**Mufei Li**[†], **Viraj Shitole**[†], **Eli Chien**[†], **Changhai Man**[†],
**Zhaodong Wang**[‡], **Srinivas Sridharan**[§], **Ying Zhang**[‡] , **Tushar Krishna**[¶], **Pan Li**[†]

[†]Georgia Institute of Technology,
{mufei.li, viraj2, ichien6, cman8, panli}@gatech.edu
[‡]Meta, {zhaodongwang, zhangying}@meta.com
[§]NVIDIA, srisridharan@nvidia.com
[¶]Georgia Institute of Technology, tushar@ece.gatech.edu

## Abstract

Directed acyclic graphs (DAGs) serve as crucial data representations in domains such as hardware synthesis and compiler/program optimization for computing systems. DAG generative models facilitate the creation of synthetic DAGs, which can be used for benchmarking computing systems while preserving intellectual property. However, generating realistic DAGs is challenging due to their inherent directional and logical dependencies. This paper introduces LayerDAG, an autoregressive diffusion model, to address these challenges. LayerDAG decouples the strong node dependencies into manageable units that can be processed sequentially. By interpreting the partial order of nodes as a sequence of bipartite graphs, LayerDAG leverages autoregressive generation to model directional dependencies and employs diffusion models to capture logical dependencies within each bipartite graph. Comparative analyses demonstrate that LayerDAG outperforms existing DAG generative models in both expressiveness and generalization, particularly for generating large-scale DAGs with up to 400 nodes—a critical scenario for system benchmarking. Extensive experiments on both synthetic and real-world flow graphs from various computing platforms show that LayerDAG generates valid DAGs with superior statistical properties and benchmarking performance. The synthetic DAGs generated by LayerDAG enhance the training of ML-based surrogate models, resulting in improved accuracy in predicting performance metrics of real-world DAGs across diverse computing platforms. Our implementation is available at https://github.com/Graph-COM/LayerDAG.

## 1 Introduction

A Directed Acyclic Graph (DAG) is a data structure that represents the order of elements (Bang-Jensen & Gutin, 2008). Unlike linear sequences, which follow a single, straight path from the first to the last element, DAGs incorporate branching and merging, allowing for the modeling of complex dependencies and hierarchies. This flexibility makes DAGs ideal for representing diverse problem domains such as workload behavior during system execution Sridharan et al. (2023), operator dependence for program analysis (Phothilimthana et al., 2023b; Luo et al., 2021; Cortez et al., 2017), dataflows in circuits (Dong et al., 2023), task dependencies in project management (Skiena, 2008; Borenstein, 2000), and cause-effect relationships (Pearl, 1995; Tennant et al., 2020). Additionally, DAGs have recently been used to create challenging benchmarks for evaluating the reasoning capabilities of large language models (Zhu et al., 2024).

Despite the benefits of DAGs mentioned above, it is non-trivial to get largescale datasets of real DAGs to enable analysis and optimization. Taking workload execution as an example, DAGs capturing execution behavior can help engineers gain valuable insights into performance metrics such

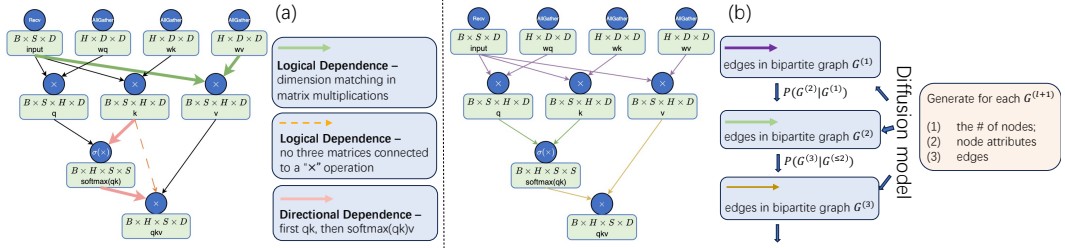

Figure 1: (a) A real-world DAG (the computation flow for a transformer layer (Vaswani et al., 2017)) encompasses complex logical and directional dependencies. Examples of logical dependencies include 1) dimension matching in matrix multiplications and 2) exactly two matrices pointed to a × operation. One example of directional dependencies here is SOFTMAX(QK)V being computed after QK. (b) Each DAG has a unique layerwise partition, an ordered partition of nodes/edges into a sequence of bipartite graphs. In LayerDAG, each bipartite graph $G^{(l+1)}$ is generated by a diffusion model conditioned on $G^{(\leq l)}$. LayerDAG generates in order the number of new nodes, their attributes, and the new edges.

as latency and resource consumption exhibited by candidate systems (Luo et al., 2021; Cortez et al., 2017) to optimize the software and hardware accordingly. However, collecting a workload execution DAG (Sridharan et al., 2023) for even a single Large Language Model training job, potentially involving trillions of operations, is extremely prohibitive given the size of modern AI platforms. For example, Meta's Llama3 was trained on a 24K GPU platform (Dubey et al., 2024) while xAI is building a 100K GPU platform. Moreover, few companies in the world even have access to platforms of this scale. Even if such a large workload execution DAG could be collected, practical constraints such as the storage requirements of the DAGs, and potential to leak private information about the AI model architecture or the training platform configuration further limits DAG sharing. Furthermore, a representative and smaller graph is more effective for most performance optimization tasks.

This work focuses on developing generative models for DAGs. We believe this can solve several challenges. First, being able to be data-driven and generate representative small set of DAGs can be much more compute and memory efficient than a real DAG given repetitive patterns. Second, this could enable data sharing without compromising intellectual property (Gao et al., 2024; Lin et al., 2020), thereby promoting software and hardware co-design among different companies (Sridharan et al., 2023). Third, a conditional DAG generative model can efficiently search the space of valid DAGs for optimization purposes in scenarios such as circuit design (Takagi, 1999; Dong et al., 2023), compiler optimization (Aho et al., 2006; Phothilimthana et al., 2023b), and neural architecture search (NAS) (Zhang et al., 2019; Li et al., 2023a; An et al., 2023).

Serving as an abstraction for flows and node dependencies, DAGs pose significant challenges in developing powerful and efficient generative models due to their intrinsic strong directional and logical dependencies, such as control flows, logic gates, and dimension requirements of matrix operations (as illustrated in Fig. 1 (a)). These complexities are further magnified in large-scale DAGs, presenting a unique combination of challenges regarding both scale and logical rules. In contrast, although social networks may be large in scale, they do not exhibit strong dependencies between nodes and edges. At the other end of the spectrum, molecular graphs, whose generation has also received much interest recently, must adhere to chemical rules but are generally smaller in size.

This work proposes the use of autoregressive diffusion models to generate DAGs, aiming to decouple the strong node dependencies in DAGs into manageable units and handle them sequentially. Our model, named LayerDAG, is based on a novel perspective of DAGs: as illustrated in Fig. 1 (b), the partial order of nodes dictated by the DAG structure can be decoupled as a sequence of tokens, each corresponding to a bipartite graph. This perspective enables the natural modeling of directional dependencies in DAGs through autoregression, and the rest is to address the logical dependencies within a set of nodes incomparable (i.e. no ordering relations) according to the partial order in each autoregressive step, where we leverage diffusion models known for effectively capturing multidimensional dependencies (Rombach et al., 2022; Vignac et al., 2023a). Specifically, the diffusion model is conditioned on previously generated bipartite graphs to generate the next bipartite graph consisting of node attributes and edges.

Our model advances existing DAG generative models in multiple aspects (Zhang et al., 2019; Li et al., 2023a; An et al., 2023). Methodologically, although autoregressive models have been adopted by D-VAE (Zhang et al., 2019) and GraphPNAS (Li et al., 2023a) for DAG generation, they treat either a single node or a node set of constant size as a token. This tokenization method imposes an order between nodes that should be incomparable in the partial order, violating the inductive bias inherent in the DAG structure. We argue that the violation may hurt the generalization capability of generative models. Moreover, D-VAE suffers from a potentially slow generation process by generating one node per iteration. Although GraphPNAS generates multiple nodes per iteration, it uses a mixture of Bernoulli distributions to model intra-set edge dependencies, which is less expressive than diffusion models (Wang et al., 2023; Pearce et al., 2023). DiffusionNAG (An et al., 2023) employs diffusion models to generate only node attributes given a DAG structure. Diffusion models have been used to generate undirected graphs (Niu et al., 2020; Jo et al., 2022; Vignac et al., 2023a), but they ignore the directional information in DAGs, while our work demonstrates the necessity of the autoregressive component in modeling directional dependencies in DAGs.

From the application perspective, all the existing works (Zhang et al., 2019; Li et al., 2023a; An et al., 2023) focus on generating small DAGs (with #nodes $\leq 24$) for NAS, while our model is capable of generating much larger flow graphs (up to $\sim 400$ nodes) for system/hardware benchmarking. Overall, our work is the first to use autoregressive diffusion models for DAG generation, aiming to take the advantages of both autoregressive models and diffusion models to model the strong dependencies commonly in DAG data.

We conduct extensive experiments to verify the effectiveness of our model. To assess the model's ability to learn strong directional and logical rules, we construct a challenging synthetic DAG dataset with injected dependencies for evaluation. Additionally, we employ three real-world datasets—computational graphs on Tensor Processing Units (TPU), flow graphs on Field Programmable Gate Arrays (FPGA), and neural architectures deployed on edge devices— for computing system benchmarking applications. Each dataset contains thousands of DAGs, with individual DAGs comprising up to hundreds of nodes. We compare the validity and statistical properties of the synthetic DAGs generated by our model with baselines. To measure benchmarking performance, we use the synthetic labeled DAGs to train surrogate machine learning (ML) models to predict TPU runtime, FPGA resource usage, and the inference latency of neural architectures for the three application scenarios. These trained surrogate models are then applied to the real-world DAGs for testing. Note that ML-based surrogate models are widely used today to measure the performance of systems and programs without the need for time-consuming simulations (Adams et al., 2019; Chen et al., 2018; Mendis et al., 2019; Sỳkora et al., 2022; Steiner et al., 2021; Baghdadi et al., 2021; Zheng et al., 2020; Li et al., 2020; Ahn et al., 2020; Dubach et al., 2007; Jia et al., 2020; Kaufman et al., 2021; Cao et al., 2023; Phothilimthana et al., 2023b). We compare the predictions given by the surrogate models with the ground-truth behavior of the DAGs on the corresponding systems. Results show that the surrogate models trained on our generated synthetic DAGs consistently outperform the ones derived with baseline generative models. We also evaluate the extrapolation and interpolation properties of LayerDAG by generating DAGs with labels out of the regime used for training, and LayerDAG demonstrates a superior generalization capability.

## 2 PRELIMINARIES AND BACKGROUND

**DAG generation** A DAG is a directed graph without cycles $G = (\mathcal{V}, \mathcal{E}, \mathbf{X})$, where $\mathcal{V} = \{1, \cdots, N\}$ is the node set, $\mathcal{E} = \{\cdots, (u,v), \cdots\}$ is the set of directed edges, $\mathbf{X}$ is the node attribute matrix. The edge list $\mathcal{E}$ can also be equivalently represented as the adjacency matrix $\mathbf{A}$. From a collection of DAGs $\{G_i\}_i \sim \mathbb{P}_{\mathcal{G}}$, we aim to learn $\mathbb{P}_{\mathcal{G}}$ with a generative model $\mathbb{P}_{\mathcal{G}}^{\theta}$, where $\theta$ denotes model parameters. Additionally, many real-world applications, especially computing system benchmarking, involve labeled DAGs $\{(G_i, y_i)\}_i$ where the labels indicate the performance of the corresponding computing systems. DAG generation conditioning on these labels is also of interest, i.e., learning the distribution $\mathbb{P}_{\mathcal{G}|\mathcal{Y}}$.

**Discrete denoising diffusion** While our framework is compatible with various diffusion models, we adopt Discrete Denoising Diffusion Probabilistic Model (D3PM) (Austin et al., 2021) as a proof of concept. We briefly review it here. D3PM (Austin et al., 2021) generalizes Multinomial Diffusion (Hoogeboom et al., 2021b) in extending DDPM (Ho et al., 2020) to discrete data. D3PM has

two phases, a forward diffusion process and a reverse process. Let each entry of $\mathbf{Z}^{(0)} \in \{0, 1\}^{M \times C}$ be a one-hot encoding of a categorical attribute with $C$ possible values.

The forward process uses $T$ consecutive steps to progressively corrupt $\mathbf{Z}^{(0)} \rightarrow \mathbf{Z}^{(1)} \rightarrow \cdots$ into purely random variables $\mathbf{Z}^{(T)}$ where $\mathbf{Z}^{(T)}$ can be drawn from a prior categorical distribution. To corrupt $\mathbf{Z}^{(t)}$ into $\mathbf{Z}^{(t+1)}$, it computes and samples from a conditional distribution $q(\mathbf{Z}^{(t+1)}|\mathbf{Z}^{(t)}, t) = \mathbf{Z}^{(t)}\mathbf{Q}^{(t+1)}$, where $\mathbf{Q}^{(t+1)} \in \mathbb{R}^{C \times C}$ is a pre-determined transition matrix. A denoising network $\phi_\theta$ is trained to predict the uncorrupted data $\mathbf{Z}^{(0)}$ from $(\mathbf{Z}^{(t)}, t)$. During the reverse generation process, the trained denoising network is used to convert $\mathbf{Z}^{(T)}$ drawn from the prior distribution into realistic data. DiGress (Vignac et al., 2023a) extends D3PM for molecular graph (undirected) generation by treating the presence or absence of an edge between a node pair as a binary attribute. The denoising of node attributes and edges takes the form of node classification and link prediction, respectively.

## 3 METHODOLOGY

In this section, we present the LayerDAG framework. We first describe our novel view of DAGs, which uniquely transforms a DAG into a sequence of bipartite graphs. This naturally leads to a layerwise tokenization for autoregressive generation, which tackles the directional dependencies in DAGs. In each autoregressive step, we introduce a layerwise diffusion model that is capable of modeling the dependencies between nodes that are incomparable (no by-default order). We also discuss why our way of DAG decomposition is crucial for model generalization. Lastly, as bipartite graphs across different layers hold different levels of complexity, we also propose a flexible quality-efficiency trade-off strategy that adjusts the number of diffusion steps based on layer index.

### 3.1 A UNIQUE SEQUENTIAL VIEW OF DAGS

The structure of a DAG essentially establishes a partial order among its nodes based on reachability through directed paths. A partially ordered set can be transformed into an ordered sequence while preserving compatibility with the partial order, known as a linear extension or a topological ordering in the context of DAGs (Kahn, 1962; Tarjan, 1972). Previous competitive autoregressive models of DAGs utilize this sequential nature to generate nodes one at a time following a topological ordering (Zhang et al., 2019). However, topological orderings are not unique. This may compromise the model's effectiveness and efficiency. We leave a dedicated discussion about the issue to Section 3.3. On the other hand, non-autoregressive models like diffusion models may be sub-optimal for not explicitly modeling the sequential nature of DAGs.

We map DAGs into a unique sequence by extending topological orderings from a sequence of single-ton subsets to general subsets. A DAG is guaranteed to have a set of source nodes, i.e., the nodes are not destinations of any edges, and we denote them by $\mathcal{V}^{(1)}$. Iteratively, we take $\mathcal{V}^{(l+1)} \subset \mathcal{V} \setminus \mathcal{V}^{(\leq l)}$ to be the set of nodes whose predecessors are in $\mathcal{V}^{(\leq l)}$, where $\mathcal{V}^{(\leq l)} = \bigcup_{i=1}^{l} \mathcal{V}^{(i)}$. An implied property is that for each $v \in \mathcal{V}^{(l+1)}$, there exists $u \in \mathcal{V}^{(l)}$ such that $(u, v) \in \mathcal{E}$. In addition, for each $u \in \mathcal{V}^{(l)}$, the longest path from source nodes to it has a length of $l - 1$. It follows that $(\mathcal{V}^{(1)}, \mathcal{V}^{(2)}, \cdots, \mathcal{V}^{(L)})$ forms an ordered partition of $\mathcal{V}$ the moment $\mathcal{V}^{(L+1)} = \emptyset$ (Fig. 1 (b)). We refer to $\mathcal{V}^{(1)}, \mathcal{V}^{(2)}, \cdots, \mathcal{V}^{(L)}$ as layers and $L$ as the number of layers. For each layer depth $1 \leq l \leq L - 1$, we also take $\mathcal{E}^{(l+1)} = \{(u, v) \in \mathcal{E} | u \in \mathcal{V}^{(\leq l)}, v \in \mathcal{V}^{(l+1)}\}$, and $(\mathcal{E}^{(2)}, \mathcal{E}^{(3)}, \cdots, \mathcal{E}^{(L)})$ forms an ordered partition of $\mathcal{E}$. This layerwise partition naturally extends to arbitrary node and edge attributes. Furthermore, this way of construction is unique and allows reconstructing the original DAG from the sequence $(\mathcal{V}^{(1)}, (\mathcal{V}^{(\leq 1)} \cup \mathcal{V}^{(2)}, \mathcal{E}^{(2)}), \cdots, (\mathcal{V}^{(\leq L-1)} \cup \mathcal{V}^{(L)}, \mathcal{E}^{(L)}))$, where $(\mathcal{V}^{(\leq l)} \cup \mathcal{V}^{(l+1)}, \mathcal{E}^{(l+1)})$ is a bipartite graph whose edges point from one part to another part. Such an invertible process of converting a raw data sample into a sequence is known as tokenization in the context of generative models, with notable examples being subwords for language models (Sennrich et al., 2016) and patches for image generation (Peebles & Xie, 2023). Essentially, layerwise partition leads to a layerwise tokenization for DAGs.

## 3.2 LAYERDAG

**Autoregressive generation** The layerwise tokenization motivates the following factorization for a probability distribution $\mathbb{P}(G)$ of DAGs. Let $G = (\mathcal{V}, \mathbf{X}, \mathbf{A})$ be a DAG. In analogy to $\mathcal{V}^{(l)}$ and $\mathcal{V}^{(\le l)}$, we define $\mathbf{X}^{(l)}$, $\mathbf{X}^{(\le l)}$, $\mathbf{A}^{(l)}$, and $\mathbf{A}^{(\le l)}$. In addition, we define $G^{(l)} = \left( \mathcal{V}^{(l)}, \mathbf{X}^{(l)}, \mathbf{A}^{(l)} \right)$ and $G^{(\le l)} = \left( \mathcal{V}^{(\le l)}, \mathbf{X}^{(\le l)}, \mathbf{A}^{(\le l)} \right)$. Then, we have

$$\mathbb{P}(G) = \prod_{l=0}^{L-1} \mathbb{P}\left( G^{(l+1)} \,\middle|\, G^{(\le l)} \right)$$

$$= \prod_{l=0}^{L-1} \mathbb{P}\left( |\mathcal{V}^{(l+1)}| \,\middle|\, G^{(\le l)} \right) \mathbb{P}\left( \mathbf{X}^{(l+1)} \,\middle|\, G^{(\le l)}, |\mathcal{V}^{(l+1)}| \right) \mathbb{P}\left( \mathbf{A}^{(l+1)} \,\middle|\, G^{(\le l)}, \mathbf{X}^{(l+1)} \right)$$

This factorization naturally leads to a layerwise autoregressive generation framework. Iteratively, to generate the $(l + 1)^{\text{th}}$ layer, it first predicts the number of new nodes with $p_\theta \left( |\mathcal{V}^{(l+1)}| \mid G^{(\le l)} \right)$. Then it generates the node attributes with $p_\theta \left( \mathbf{X}^{(l+1)} \mid G^{(\le l)}, |\mathcal{V}^{(l+1)}| \right)$. Finally, it generates the edges with $p_\theta \left( \mathbf{A}^{(l+1)} \mid G^{(\le l)}, \mathbf{X}^{(l+1)} \right)$. The generation process terminates when $|\mathcal{V}^{(l+1)}|$ is predicted to be 0. All the above generation processes will be conditioned on the previously generated DAGs $G^{(\le l)}$. With layerwise autoregressive generation, this framework potentially allows better generalization to unseen values of $L$ during generation (layer generalization).

**Layerwise diffusion-based generation** To preserve the uniqueness of the layerwise tokenization, we need to generate the set of node attributes and edges in the bipartite graph $G^{(l+1)}$ as a whole by modeling $p_\theta \left( \mathbf{X}^{(l+1)} \mid G^{(\le l)}, |\mathcal{V}^{(l+1)}| \right)$ and $p_\theta \left( \mathbf{A}^{(l+1)} \mid G^{(\le l)}, \mathbf{X}^{(l+1)} \right)$ as set-level generations. This requires capturing complex dependencies between the node attributes and edges in the sets, which is crucial for applications like system and circuit design, where strict logical rules are prevalent, and rule violations can accumulate and propagate over layers of generation. To tackle this issue, we adopt diffusion models for multiple rounds of set refinement in the generation of $G^{(l+1)}$. We employ two separate diffusion processes for node attribute $\mathbf{X}^{(l+1)}$ generation and structure $\mathbf{A}^{(l+1)}$ generation, as suggested by Li et al. (Li et al., 2023b). For simplicity, we focus on categorical node attributes in this work, which find abundant real-world applications like operator types in computational graphs, and thus adopt D3PM (Austin et al., 2021). Our approach can be extended to handle real-valued attributes by combining discrete diffusion with continuous diffusion, as demonstrated in previous works (Vignac et al., 2023b; Hua et al., 2024). For more details on layerwise diffusion, see Appendix A.

**Implementation** For different $l$'s, all modules $p_\theta \left( |\mathcal{V}^{(l+1)}| \mid G^{(\le l)} \right)$, $p_\theta \left( \mathbf{X}^{(l+1)} \mid G^{(\le l)}, |\mathcal{V}^{(l+1)}| \right)$, and $p_\theta \left( \mathbf{A}^{(l+1)} \mid G^{(\le l)}, \mathbf{X}^{(l+1)} \right)$ share the same parameter $\theta$, which involves a DAG encoder. We use an off-the-shelf bidirectional message passing neural network (BiMPNN) (Wen et al., 2020). A single BiMPNN layer updates node representations with synchronous message passing over both the directed edges and their reversed counterparts: $\sigma \left( \mathbf{A} \mathbf{H} \mathbf{W}_{\text{forward}} + \mathbf{A}^\top \mathbf{H} \mathbf{W}_{\text{reverse}} + \mathbf{H} \mathbf{W}_{\text{self}} \right)$, where $\sigma$ is a non-linear layer, $\mathbf{H}$ is the node representation matrix, and $\mathbf{W}$'s are learnable weight matrices. Both layer size generation $p_\theta \left( |\mathcal{V}^{(l+1)}| \mid G^{(\le l)} \right)$ and node attribute generation $p_\theta \left( \mathbf{X}^{(l+1)} \mid G^{(\le l)}, |\mathcal{V}^{(l+1)}| \right)$ involve computing graph representations with a set pooling operator over the updated node representations. Let $\mathbf{X}^{(l+1,t)}$ be the node attributes sampled for the $(l + 1)^{\text{th}}$ layer at the step $t$ in the reverse node attribute diffusion process. After encoding $(G^{(\le l)}, t)$ into a context vector $\mathbf{h}_t^{(\le l)}$, the denoising network $\phi_{\theta_X}$ augments the representations of $\mathbf{X}^{(l+1,t)}$ by $\mathbf{h}_t^{(\le l)}$ and then applies a transformer without positional encodings (Vaswani et al., 2017) over them for predicting the set of $\mathbf{X}^{(l+1)}$. Similarly, the denoising network $\phi_{\theta_A}$ for edge diffusion augments $(G^{(\le l)}, t)$ by $(\mathbf{X}^{(l+1)}, \mathbf{A}^{(l+1,t)})$ for computing the node representations of $\mathcal{V}^{(\le l+1)}$. To predict the probability of edge $(u, v)$ for $u \in \mathcal{V}^{(\le l)}$ and $v \in \mathcal{V}^{(l+1)}$, it concatenates and transforms the representations of node $u$, node $v$, and $t$ with an MLP.

**Training** The training process aims to maximize the log-likelihood of observed graphs under our model. We decompose this objective to train the three modules independently. To account for the autoregressive nature of our approach, we train all modules with teacher forcing. In each training iteration, we randomly sample a real partial DAG up to $l$ layers and train the model to predict the $l +$

1-th layer. For a given partial DAG, the training of $p_\theta \left( |\mathcal{V}^{(l+1)}| \mid G^{(\leq l)} \right)$ is analogous to the standard supervised training of a graph classification model. The training of $p_\theta \left( \mathbf{X}^{(l+1)} \mid G^{(\leq l)}, |\mathcal{V}^{(l+1)}| \right)$ and $p_\theta \left( \mathbf{A}^{(l+1)} \mid G^{(\leq l)}, \mathbf{X}^{(l+1)} \right)$ follows the established practices for discrete diffusion models, where the modules are optimized by recovering corrupted discrete data.

**Sampling** The sampling process follows an autoregressive approach, performing layerwise graph generation. Starting with an empty graph $G^{(\leq 0)}$, we iteratively sample the number of nodes $|\mathcal{V}^{(l+1)}|$ for each subsequent layer using $p_\theta \left( |\mathcal{V}^{(l+1)}| \mid G^{(\leq l)} \right)$. If $|\mathcal{V}^{(l+1)}|$ is non-zero, we sequentially generate node attributes $\mathbf{X}^{(l+1)}$ and edges $\mathbf{A}^{(l+1)}$ using their respective reverse diffusion processes. We then update $G^{(\leq l+1)}$ with the new nodes, attributes, and edges. This process continues until we predict $|V^{(l+1)}|$ to be 0, at which point generation terminates.

**Conditional generation** LayerDAG is also capable of performing conditional generation. Given a labeled DAG $(G, y)$, we train LayerDAG to learn the conditional distribution $\mathbb{P}(G|Y)$ by integrating the sinusoidal embeddings of $y$ into the representations of $\mathbf{X}^{(\leq l)}$ for any $l$. Once trained, the model can generate DAGs conditioned on specified target properties.

### 3.3 Permutation invariance and model generalization

A critical issue in developing probabilistic generative models of graphs is permutation invariance, whether the probability for a model to generate a graph is invariant to the particular choice of node ordering. Non-permutation invariant models, such as autoregressive models that generate one node at a time, require data augmentations with random node orderings during training (You et al., 2018; Li et al., 2018). To sufficiently train the model, one has to enumerate a large number of orderings, which could be exponential in $N$. LayerDAG is permutation invariant with the aforementioned implementation. By aligning with such an inductive bias, LayerDAG has the potential of holding a good generalization capability with limited computational resources for training.

**Proposition 3.1 (permutation invariance of LayerDAG)** *For any depth l, $p_\theta \left( |\mathcal{V}^{(l+1)}| \mid G^{(\leq l)} \right)$, $p_\theta \left( \mathbf{X}^{(l+1)} \mid G^{(\leq l)}, |\mathcal{V}^{(l+1)}| \right)$, and $p_\theta \left( \mathbf{A}^{(l+1)} \mid G^{(\leq l)}, \mathbf{X}^{(l+1)} \right)$ are permutation invariant. Hence, LayerDAG is permutation invariant.*

Bidirectional MPNN layers are permutation equivariant, and sum and mean pooling operators are permutation invariant. It follows that for any permutation $\Pi$, we have $\mathbb{P}\left( \Pi^{(l+1)}(G^{(l+1)}) \mid \Pi^{(\leq l)}(G^{(\leq l)}) \right) = \mathbb{P}\left( G^{(l+1)} \mid G^{(\leq l)} \right)$, where $\Pi^{(l+1)}$ corresponds to $\mathcal{V}^{(l+1)}$ and $\Pi^{(\leq l)}$ corresponds to $\mathcal{V}^{(\leq l)}$. Finally, $\mathbb{P}(G) = \prod_{l=0}^{L-1} \mathbb{P}\left( G^{(l+1)} \mid G^{(\leq l)} \right)$ is also permutation invariant.

### 3.4 Flexible quality-efficiency trade-off with a layer-index-based denoising schedule

Note that as $l$ increases, both $|\mathcal{V}^{(\leq l)}|$ and $|\mathcal{E}^{(l+1)}|$ increase in general, resulting in more complex edge dependencies. To effectively handle this pattern, we introduce a non-uniform denoising schedule for better allocation of the time budget. Specifically, we propose to set the total number of denoising steps for a layer to linearly increase in $l$.

$$T^{(l)} = T_{\min} + \lfloor (T_{\max} - T_{\min}) \cdot \min \{l/L_{\max}, 1\} \rfloor$$

where $T_{\min} \leq T_{\max}$ are the minimum and maximum number of denoising steps and allow users to make a flexible quality-efficiency trade-off for DAG generation with LayerDAG. $l$ is the current layer index, $L_{\max}$ is the maximum number of layers in the training data, and $\lfloor \cdot \rfloor$ is the floor function.

The key assumption is that more complex patterns emerge as the layer depth increases, such as long range dependencies between a pair of layers with a big difference in layer depth, which calls for the use of more denoising steps. When this does not hold, as in the scenario of trees with stationary layer distributions, this layer-based linear denoising schedule will not be useful.

## 4 RELATED WORK

We have already extensively compared LayerDAG with DAG generative models. Here, we review relevant undirected graph generative models.

**Autoregressive models.** GraphRNN (You et al., 2018) and DeepGMG (Li et al., 2018) are node-wise autoregressive models for undirected graphs, generating one node at a time. D-VAE extends this approach for DAGs and adopts topological orderings (Zhang et al., 2019). For efficiency considerations, GRAN proposes sequentially generating node sets of constant size (Liao et al., 2019). GraphPNAS extends this approach for attributed DAG generation (Li et al., 2023a). Like their DAG counterparts, GraphRNN and GRAN also violate the inductive bias by imposing an order of the nodes. Consequently, they may suffer from inferior effectiveness or training efficiency.

**Diffusion models.** Niu et al. applies continuous diffusion models to adjacency matrices of undirected graphs (Niu et al., 2020). GDSS extends the previous approach to attributed graphs like molecules (Jo et al., 2022). DiGress proposes a discrete diffusion model for attributed graph generation (Vignac et al., 2023a). GraphMaker extends DiGress for generating large attributed graphs (Li et al., 2023b). All these models use a constant number of denoising steps in the generation of the whole graph, regardless of their size and complexities. Recently, there has been an arising interest in exploring the intersections of autoregressive models and diffusion models. TimeGrad (Rasul et al., 2021) proposes an autoregressive diffusion model for time series forecasting. ARDMs (Hoogeboom et al., 2021a) generalize order-agnostic autoregressive models and absorbing discrete diffusion. For undirected graph generation, EDGE (Chen et al., 2023) and GRAPHARM (Kong et al., 2023) propose diffusion models where each denoising step generates a new set of edges and therefore also corresponds to exactly one step in autoregressive generation. Unlike LayerDAG, they do not perform multiple rounds of refinement, which is critical for capturing logical dependencies among node attributes and edges. In contrast, with autoregressive layerwise diffusion, LayerDAG allows for multiple-step refinement in each layer generation, and with the layer-index-based denoising schedule, LayerDAG also allows for using more denoising steps for more complex layers, which provides the overall freedom to balance quality and efficiency.

## 5 EXPERIMENTS

The end application scenarios of DAG generative models pose multifaceted requirements that motivate the design of our empirical studies. **Q1**) Given that real-world DAGs encompass complex logical and directional dependencies, where violations can directly invalidate generated samples, how effectively can LayerDAG capture these dependencies by learning from valid DAGs? **Q2**) Beyond ensuring validity, the application of synthetic data for system and hardware benchmarking necessitates the preservation of correlations between DAGs and system metrics such as throughput, latency, and memory footprint. How effectively can LayerDAG perform DAG generation conditioning on these metrics to meet these requirements? **Q3**) With the rapid advancements in computational workloads, including foundation models and edge computing, system vendors need to proactively design next-generation platforms. A model capable of simulating workloads with hypothetical metrics (e.g. low latency / low power consumption) holds significant potential. Additionally, optimizing DAG properties for scenarios like circuit design demands generalization to unseen property values. Thus arises a crucial question: can LayerDAG generalize to unseen regimes of DAG labels when generating DAGs? **Q4**) Different application scenarios require varying trade-offs between generation efficiency and quality. How effectively does a layer-index-based denoising schedule (in Sec. 3.4) meet this requirement using a single trained LayerDAG model?

**Baselines** We consider three non-diffusion autoregressive baselines – **GraphRNN**, **D-VAE**, and **GraphPNAS**. GraphRNN originally tackles undirected graph structure generation You et al. (2018), and we adopt an extension of it for attributed DAG generation Zhang et al. (2019). D-VAE is a variational auto-encoder and employs a nodewise autoregressive decoder. Both GraphRNN and D-VAE employ topological orderings and one-hot encoding of node IDs Zhang et al. (2019). GraphPNAS sequentially generates incident edges for a node set of constant size, using a mixture of Bernoulli distributions to model intra-set dependencies. To compare the capability of LayerDAG in set generation against GraphPNAS, we further adapt GraphPNAS by using mixtures of multinoulli distributions for generating a set of node attributes and setting the node set size to be the averaged layer size. For

Table 1: Evaluation results on LP. Best results are in **bold**.

| Model | $\rho = 0$ | | | $\rho = 0.5$ | | | $\rho = 1$ | | |
|---|---|---|---|---|---|---|---|---|---|
| | Validity $\uparrow$ | $W_1$ / MMD $\downarrow$ | | Validity $\uparrow$ | $W_1$ / MMD $\downarrow$ | | Validity $\uparrow$ | $W_1$ / MMD $\downarrow$ | |
| | | $L (\times 10^{-1})$ | $|\mathcal{V}^{(l)}| (\times 10^{-1})$ | | $L$ | $|\mathcal{V}^{(l)}|$ | | $L$ | $|\mathcal{V}^{(l)}|$ |
| D-VAE | $0.27 \pm 0.03$ | $8.7 \pm 1.0$ | $1.9 \pm 0.3$ | $0.37 \pm 0.04$ | $9.8 \pm 1.6$ | $1.9 \pm 0.6$ | $0.89 \pm 0.01$ | $8.8 \pm 0.9$ | $1.9 \pm 0.5$ |
| GraphRNN | $0.25 \pm 0.02$ | $9.8 \pm 0.2$ | $1.2 \pm 0.2$ | $0.34 \pm 0.07$ | $12.0 \pm 0.0$ | $1.8 \pm 0.2$ | $0.59 \pm 0.02$ | $14.0 \pm 1.0$ | $2.1 \pm 0.1$ |
| GraphPNAS | $0.23 \pm 0.04$ | $17.0 \pm 4.0$ | $2.2 \pm 0.7$ | $0.24 \pm 0.03$ | $20.0 \pm 3.0$ | $3.2 \pm 1.3$ | $0.67 \pm 0.04$ | $10.0 \pm 3.0$ | $0.8 \pm 0.6$ |
| OneShotDAG | $0.37 \pm 0.02$ | $6.4 \pm 0.9$ | $1.5 \pm 0.1$ | $0.31 \pm 0.07$ | $3.9 \pm 0.7$ | $1.3 \pm 0.0$ | $0.50 \pm 0.08$ | $4.1 \pm 2.4$ | $1.1 \pm 0.4$ |
| LayerDAG ($T = 1$) | $0.26 \pm 0.06$ | $\mathbf{1.6 \pm 0.8}$ | $\mathbf{0.14 \pm 0.0}$ | $0.36 \pm 0.02$ | $\mathbf{1.3 \pm 0.3}$ | $\mathbf{0.12 \pm 0.1}$ | $0.95 \pm 0.01$ | $\mathbf{2.0 \pm 0.1}$ | $\mathbf{0.08 \pm 0.0}$ |
| LayerDAG | $\mathbf{0.56 \pm 0.02}$ | $\mathbf{1.6 \pm 1.0}$ | $\mathbf{0.10 \pm 0.0}$ | $\mathbf{0.63 \pm 0.00}$ | $\mathbf{1.8 \pm 1.1}$ | $\mathbf{0.06 \pm 0.0}$ | $\mathbf{0.96 \pm 0.02}$ | $\mathbf{1.9 \pm 0.6}$ | $\mathbf{0.10 \pm 0.3}$ |

pure diffusion baselines, we implement **OneShotDAG**, a non-autoregressive variant of LayerDAG. For the ablation study of multiple rounds of refinement, we report results with a single denoising step, denoted by **LayerDAG (T = 1)**. For details of model extensions, see Appendix B.

## 5.1 GENERATING SYNTHETIC DAGS WITH STRONG LOGICAL RULES (**Q1**)

To evaluate the model's capability in capturing logical rules by learning from valid DAGs, we propose a synthetic dataset of latent preferential DAGs (**LP**). LP adheres to various hard logical constraints, including a constraint on the balanced level of binary node attributes among the node's predecessors. Specifically, we enforce that $\frac{\lfloor |n_v^{(0)} - n_v^{(1)}|/2 \rfloor}{(n_v^{(0)} + n_v^{(1)})/2} \leq \rho$ for any node $v$, where $\lfloor \cdot \rfloor$ is the floor function, and $n_v^{(i)}$ is the number of node $v$'s predecessors with attribute $i$ for $i \in \{0, 1\}$. The parameter $\rho \in \{0, 0.5, 1\}$ helps assess how model performance varies under different degrees of constraint, where a lower value imposes stricter constraints ($\rho = 0$ indicating $|n_v^{(0)} - n_v^{(1)}| \leq 1$). For a full description of the dataset, see Appendix C.

We use LP with different $\rho$'s to generate the datasets $D_\rho$ and train different generative models based on $D_\rho$. After training, we use each model to generate DAGs of the same number as that in $D_\rho$. To evaluate the hard logical constraints, we assess the validity of generated DAGs by measuring the proportion of generated DAGs that satisfy the imposed hard constraints. To evaluate the soft constraints, we compare the distributions of graph statistics between generated DAGs and an equal number of real DAGs. We measure the 1-Wasserstein distance ($W_1$) between the distributions of layer numbers ($L$) for the two graph sets. Additionally, we measure Maximum Mean Discrepancy (MMD) You et al. (2018) between two sets of graph statistic distributions corresponding to the graph sets. Specifically, we report MMD for the distributions of layer size ($|\mathcal{V}^{(l)}|$).

Table 1 presents the evaluation results. LayerDAG consistently outperforms all the other models in terms of validity, with substantial margins observed under stricter logical rules (lower $\rho$ values), about 20% in absolute value. Nodewise autoregressive models (GraphRNN and D-VAE) that directly encode node IDs struggle to learn strict logical rules. Meanwhile, a mixture of Bernoulli/multinoulli distribution is also not expressive enough, resulting in GraphPNAS achieving low validity scores. The ablation studies against the non-autoregressive variant (OneShotDAG) and the single-denoising-step variant ($T = 1$) underscore the importance of combining autoregressive layerwise generation and diffusion in modeling strong directional and logical rules for DAG generation. In terms of graph statistics, the layerwise autoregressive models, LayerDAG ($T = 1$) and LayerDAG, yield a better performance in capturing layerwise patterns ($L$ and $|\mathcal{V}^{(l)}|$), demonstrating the benefit of autoregressive layerwise generation. We also experiment with a variant of the LP dataset that allows directly examining the validity of the generated node attributes, where LayerDAG also achieves the best performance consistently. See Appendix D for more details.

## 5.2 CONDITIONAL GENERATION FOR REAL-WORLD DATASETS FROM DIVERSE COMPUTATIONAL PLATFORMS (**Q2**)

**Datasets.** We repurpose three representative real-world datasets, originally developed for DAG property prediction, to serve as testbeds for conditional DAG generation. The datasets are associated with computation workloads executed on diverse hardware platforms, and they well fit the end scenario of synthetic data sharing for system/hardware benchmarking. Originally released as part of the TpuGraphs dataset, **TPU Tile** is a collection of kernel graphs for machine learning workload

Table 2: Dataset statistics. $|\mathcal{V}|$, $|\mathcal{E}|$, and $L$ are averaged over graphs. $|\mathcal{V}^{(l)}|$ is averaged over layers.

| Dataset | # graphs | $|\mathcal{V}|$ | max $|\mathcal{V}|$ | $|\mathcal{E}|$ | max $|\mathcal{E}|$ | $L$ | max $L$ | $|\mathcal{V}^{(l)}|$ | max $|\mathcal{V}^{(l)}|$ | # attributes | label info |
|---|---|---|---|---|---|---|---|---|---|---|---|
| TPU Tile | 6, 301 | 40.8 | 394 | 42.9 | 711 | 11.2 | 72 | 3.6 | 21 | 1 | TPU runtime |
| HLS | 2, 062 | 88.6 | 356 | 110.7 | 477 | 27.75 | 78 | 3.2 | 28 | 7 | FPGA resource usage |
| NA-Edge | 2, 000 | 231.1 | 339 | 265.8 | 385 | 149.1 | 185 | 1.5 | 4 | 14 | mobile CPU inference latency |

Table 3: Evaluation results for conditional generation. Best results are in **bold**.

| Model | TPU Tile | | | | HLS | | | | NA-Edge | | | |
|---|---|---|---|---|---|---|---|---|---|---|---|---|
| | ML | | $W_1$ / MMD $\downarrow$ | | ML | | $W_1$ / MMD $\downarrow$ | | ML | | $W_1$ / MMD $\downarrow$ | |
| | Pearson | MAE | $L$ | $|\mathcal{V}^{(l)}|(\times 10^{-1})$ | Pearson | MAE | $L$ | $|\mathcal{V}^{(l)}|(\times 10^{-1})$ | Pearson | MAE | $L(\times 10)$ | $|\mathcal{V}^{(l)}|(\times 10^{-1})$ |
| Real graphs | $0.75 \pm 0.01$ | $0.9 \pm 0.0$ | | | $0.98 \pm 0.00$ | $0.3 \pm 0.0$ | | | $0.996 \pm 0.000$ | $0.3 \pm 0.0$ | | |
| D-VAE | $0.50 \pm 0.01$ | $1.4 \pm 0.0$ | $2.6 \pm 0.1$ | $1.3 \pm 0.4$ | $0.82 \pm 0.04$ | $1.2 \pm 0.1$ | $\underline{\mathbf{3.2 \pm 1.7}}$ | $1.5 \pm 0.2$ | $0.877 \pm 0.026$ | $2.3 \pm 0.8$ | $3.6 \pm 0.7$ | $7.3 \pm 1.3$ |
| GraphRNN | $0.62 \pm 0.02$ | $1.3 \pm 0.0$ | $1.9 \pm 0.2$ | $0.4 \pm 0.1$ | $0.79 \pm 0.03$ | $\underline{1.1 \pm 0.1}$ | $11 \pm 1.0$ | $2.4 \pm 0.3$ | $0.980 \pm 0.010$ | $1.0 \pm 0.1$ | $13 \pm 2.0$ | $14 \pm 4.0$ |
| GraphPNAS | $0.24 \pm 0.10$ | $2.1 \pm 0.6$ | $6.2 \pm 0.5$ | $1.0 \pm 0.3$ | $0.66 \pm 0.05$ | $\underline{2.5 \pm 0.6}$ | $26 \pm 0.0$ | $6.8 \pm 0.1$ | $0.619 \pm 0.118$ | $7.7 \pm 2.6$ | $15 \pm 0.0$ | $14 \pm 0.0$ |
| OneShotDAG | $0.56 \pm 0.02$ | $1.4 \pm 0.1$ | $6.9 \pm 0.2$ | $3.5 \pm 0.1$ | $0.73 \pm 0.03$ | $1.4 \pm 0.1$ | $21 \pm 0.0$ | $4.4 \pm 0.1$ | $0.887 \pm 0.038$ | $3.4 \pm 1.0$ | $14 \pm 0.0$ | $9.2 \pm 0.0$ |
| LayerDAG ($T = 1$) | $0.37 \pm 0.11$ | $2.0 \pm 0.4$ | $2.0 \pm 0.4$ | $2.1 \pm 0.2$ | $0.27 \pm 0.26$ | $2.1 \pm 0.2$ | $7.9 \pm 2.2$ | $5.2 \pm 0.2$ | $0.956 \pm 0.011$ | $3.1 \pm 2.0$ | $6.1 \pm 1.6$ | $4.7 \pm 4.5$ |
| LayerDAG | $\underline{\mathbf{0.65 \pm 0.01}}$ | $\underline{\mathbf{1.2 \pm 0.1}}$ | $\underline{\mathbf{1.3 \pm 0.4}}$ | $\underline{\mathbf{0.1 \pm 0.0}}$ | $\underline{\mathbf{0.85 \pm 0.02}}$ | $\underline{1.1 \pm 0.2}$ | $11 \pm 3.0$ | $\underline{\mathbf{1.4 \pm 0.0}}$ | $\underline{\mathbf{0.990 \pm 0.005}}$ | $\underline{\mathbf{0.9 \pm 0.3}}$ | $\underline{\mathbf{1.3 \pm 0.2}}$ | $\underline{\mathbf{0.4 \pm 0.1}}$ |

on Tensor Processing Units (TPUs), with graph labels $y$ indicating the runtime averaged over a set of compilation configurations (Phothilimthana et al., 2023b). **High-level synthesis (HLS)** is a collection of data flow intermediate representation graphs for compiled C programs, with each DAG labeled according to the resource usage of look up table measured on Field Programmable Gate Arrays (FPGAs) (Wu et al., 2022). **NA-Edge** is a collection of DAGs representing neural architectures, with labels indicating their inference latency on mobile CPU (Dong & Yang, 2020; Zhang et al., 2021). Table 2 presents the dataset statistics. We perform a random train/val/test split for all datasets. For more details on dataset adaptation and data statistic distributions, see Appendix E.

**Evaluation.** Performing ground truth evaluations for conditional generation of DAGs in system and hardware design requires direct measurements on specific computational platforms. For example, the HLS dataset requires program implementation and measurement on FPGAs (Wu et al., 2022). Such evaluations are computationally costly or infeasible due to limited access. Additionally, they demand specialized domain knowledge that often exceeds the expertise of general machine learning practitioners. Recently, employing ML-based surrogate cost models has emerged as a popular and effective alternative to direct measurement in various system and hardware optimizations (Chen et al., 2018; Jia et al., 2020; Phothilimthana et al., 2023b; Kaufman et al., 2021; Mendis et al., 2019; Sỳkora et al., 2022; Steiner et al., 2021; Baghdadi et al., 2021; Zheng et al., 2020; Dubach et al., 2007). In light of these achievements, we propose to evaluate the quality of generated DAGs with ML-based surrogate models. Specifically, we partition the real labeled DAG datasets into training/validation/test subsets. Then, we use the real training and validation labels as conditions for DAG generation. The generated labeled DAGs essentially form synthetic training and validation subsets. Inspired by previous practices (Yoon et al., 2023; Li et al., 2023b), we train two ML models with BiMPNN using the same automated pipeline respectively on the real and synthetic training/validation subsets. We then compare the performance of the two models on the real test set. A generative model is considered better if its corresponding model achieves a performance closer to that of the model trained on the real subsets.

Table 3 presents the evaluation results. For a comprehensive analysis, we report two metrics for ML-based evaluation – Pearson correlation coefficient that compares the relative label differences of DAGs in the test set based on the predicted labels and the ground-truth labels, and mean absolute error (MAE) that compares the absolute difference between the predicted labels and the ground-truth labels. LayerDAG consistently achieves the best performance in ML-based evaluation. Furthermore, we assess discrepancies in graph statistics between the real validation subset and a synthetic validation subset via the same metrics used in Sec.5.1, where LayerDAG also achieves the best performance in general. Overall, these evaluation results are aligned with previous observations made for the LP dataset.

## 5.3 LABEL GENERALIZATION IN CONDITIONAL GENERATION (**Q3**)

To assess the model generalization to unseen regimes of label values, we establish a challenging out-of-distribution testbed using the TPU Tile dataset. We sort the real-world DAGs based on their labels and then partition them into five quantiles. For an experiment, we exclude one quantile of data during the development of the generative models. After training the generative models, we

Table 4: Evaluation results for label generalization. Best results are in **bold**.

| Model | 5th quantile (extrapolation) | | | | | | 4th quantile (interpolation) | | | | | |
|---|---|---|---|---|---|---|---|---|---|---|---|---|
| | BiMPNN | | Kaggle | | $W_1$ / MMD $\downarrow$ | | BiMPNN | | Kaggle | | $W_1$ / MMD $\downarrow$ | |
| | Pearson | MAE | Pearson | MAE | $L$ | $|\mathcal{V}^{(l)}|(\times10^{-1})$ | Pearson | MAE | Pearson | MAE | $L$ | $|\mathcal{V}^{(l)}|(\times10^{-1})$ |
| Real graphs | $0.81 \pm 0.01$ | $1.3 \pm 0.0$ | $0.82 \pm 0.01$ | $1.3 \pm 0.0$ | | | $0.26 \pm 0.03$ | $0.3 \pm 0.0$ | $0.31 \pm 0.00$ | $0.3 \pm 0.0$ | | |
| D-VAE | $-0.06 \pm 0.04$ | $4.1 \pm 1.0$ | $-0.09 \pm 0.02$ | $3.8 \pm 1.0$ | $3.5 \pm 2.1$ | $1.0 \pm 0.3$ | $0.05 \pm 0.05$ | $1.3 \pm 1.1$ | $-0.04 \pm 0.07$ | $0.5 \pm 0.0$ | $\underline{\mathbf{1.5 \pm 0.3}}$ | $1.3 \pm 0.5$ |
| GraphRNN | $-0.05 \pm 0.02$ | $2.3 \pm 0.2$ | $-0.01 \pm 0.09$ | $2.2 \pm 0.1$ | $2.0 \pm 0.3$ | $1.1 \pm 0.2$ | $0.03 \pm 0.11$ | $1.8 \pm 1.8$ | $0.06 \pm 0.05$ | $0.6 \pm 0.1$ | $2.4 \pm 0.2$ | $0.8 \pm 0.1$ |
| GraphPNAS | $0.02 \pm 0.12$ | $5.4 \pm 1.3$ | $-0.01 \pm 0.08$ | $4.3 \pm 1.2$ | $7.4 \pm 0.6$ | $1.8 \pm 0.3$ | $0.12 \pm 0.02$ | $3.4 \pm 1.1$ | $0.07 \pm 0.01$ | $2.5 \pm 0.6$ | $7.7 \pm 0.0$ | $1.5 \pm 0.0$ |
| OneShotDAG | $-0.11 \pm 0.02$ | $3.5 \pm 0.8$ | $-0.14 \pm 0.03$ | $2.6 \pm 0.2$ | $7.0 \pm 0.3$ | $4.1 \pm 0.3$ | $0.08 \pm 0.01$ | $0.6 \pm 0.0$ | $0.09 \pm 0.02$ | $1.5 \pm 0.2$ | $7.4 \pm 0.2$ | $4.3 \pm 0.1$ |
| LayerDAG ($T = 1$) | $0.00 \pm 0.02$ | $5.5 \pm 4.3$ | $-0.08 \pm 0.01$ | $6.4 \pm 1.5$ | $2.4 \pm 0.5$ | $2.6 \pm 0.1$ | $-0.03 \pm 0.04$ | $0.7 \pm 0.2$ | $0.07 \pm 0.01$ | $2.3 \pm 0.4$ | $2.8 \pm 0.4$ | $3.2 \pm 0.4$ |
| LayerDAG | $\mathbf{0.22 \pm 0.11}$ | $\mathbf{1.8 \pm 0.0}$ | $\mathbf{0.18 \pm 0.09}$ | $\mathbf{1.9 \pm 0.0}$ | $\mathbf{1.4 \pm 0.2}$ | $\mathbf{0.2 \pm 0.0}$ | $\mathbf{0.19 \pm 0.03}$ | $\mathbf{0.4 \pm 0.0}$ | $\mathbf{0.14 \pm 0.06}$ | $\mathbf{0.4 \pm 0.0}$ | $1.8 \pm 0.4$ | $\mathbf{0.3 \pm 0.1}$ |

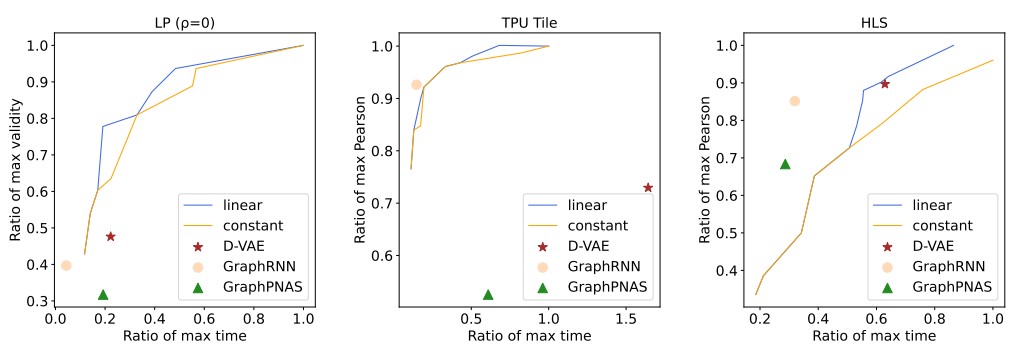

Figure 2: Generation quality with respect to time budget for LP ($\rho = 0$), TPU Tile, and HLS.

conduct a random train/val/test split on the previously excluded quantile. We evaluate conditional generation exclusively on this quantile, i.e., generating DAGs conditioning on labels from this quantile. Essentially, this setup requires the models to perform interpolation for intermediate quantiles and extrapolation for the end quantiles. An analysis of the TPU tile dataset reveals a long-tailed label distribution, with larger value ranges in latter quantiles, detailed in Appendix E. Therefore, we choose the 5th quantile for an extrapolation study and the 4th quantile for an interpolation study, which presents significant challenges. Besides the previously employed BiMPNN model, we leverage another surrogate model that stood out as a top-5 solution from more than 600 submissions in a Kaggle competition for the TpuGraphs dataset (Phothilimthana et al., 2023a; Kag, 2023).

Table 4 presents the evaluation results. As expected, the extrapolation setting is particularly difficult, with most baselines failing to attain a single positive Pearson correlation. By contrast, LayerDAG consistently outperforms competitors, improving BiMPNN's correlation by up to 0.2. While 0.2 remains modest for practical usage, the combination of autoregressive layerwise generation and diffusion drives LayerDAG 's stronger generalization across both settings.

### 5.4 EVALUATION FOR THE LAYER-INDEX-BASED DENOISING SCHEDULE (**Q4**)

Figure 2 presents the evaluation of layer-index-based denoising schedule in quality-efficiency trade-off on LP ($\rho = 0$), TPU Tile, and HLS. The curves start from the time budget that leads to the best performance. For an ablation study, we also experiment with a constant denoising schedule that simply utilizes a constant number of denoising steps $T$ for all layers. Both schedules allow an effective quality-efficiency trade-off, while the layer-index-based schedule often yields a better generation quality with the same time budget. For comparison, we also evaluate the generation time of the baselines using the same batch size. Using layer-index-based linear denoising schedule, LayerDAG exhibits a better or comparable trade-off than most baselines except GraphRNN.

## 6 CONCLUSION

We propose LayerDAG, a layerwise autoregressive diffusion model for DAGs, conceptualized by viewing a DAG as a sequence of bipartite graphs. Extensive experiments on synthetic and real-world datasets demonstrate a superior capability of LayerDAG in modeling the strong dependencies common in DAG data and in generating DAGs even with out-of-distribution labels.

ACKNOWLEDGEMENTS

We are deeply grateful to Haoyu Wang and Yinan Huang for their insights on dataset adoption and adaptation. M. Li, V. Shitole, E. Chien and P. Li are partially supported by NSF awards PHY-2117997, IIS-2239565, IIS-2428777, and CCF-2402816; DOE award DE-FOA-0002785; JPMC faculty awards; Microsoft Azure Research Credits for Generative AI; and Meta research award.

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

## A  DETAILS ON LAYERWISE DIFFUSION

Composing the transition matrices across multiple time steps yields the closed-form expression $q(\mathbf{Z}^{(t+1)}|\mathbf{Z}^{(0)}, t) = \mathbf{Z}^{(0)}\overline{\mathbf{Q}}^{(t+1)}$, where $\overline{\mathbf{Q}}^{(t+1)} = \mathbf{Q}^{(1)}\mathbf{Q}^{(2)}\cdots\mathbf{Q}^{(t+1)}$, which allows parallel training across samples and time steps. An instantiation of $\{\overline{\mathbf{Q}}^{(t)}\}_t$ is valid as long as $\lim_{t\to T}\overline{\mathbf{Q}}^{(t)}$ is a known prior distribution. During the reverse generation process, the trained denoising network is used to convert $\mathbf{Z}^{(T)}$ drawn from the prior distribution into realistic data. Iteratively, we compute and sample from $p_\theta(\mathbf{Z}^{(t-1)}|\mathbf{Z}^{(t)}, t) \propto \mathbf{Z}^{(t)}(\mathbf{Q}^{(t)})^\top \odot \hat{\mathbf{Z}}^{(0)}\overline{\mathbf{Q}}^{(t-1)}$, where $^\top$ denotes transpose, and $\odot$ denotes element-wise product.

Following the practice of DiGress, we use the empirical marginal distribution of a categorical attribute $\mathbf{m} \in \mathbb{R}^C$ as its corresponding prior distribution, which was observed to yield more efficient generation compared with a uniform prior in practice. The composed transition matrix is chosen to be $\overline{\mathbf{Q}}^{(t)} = \overline{\alpha}^{(t)}\mathbf{I} + (1 - \overline{\alpha}^{(t)})\,\mathbf{1}\mathbf{m}^\top$, where $\overline{\alpha}^{(t)} = \cos^2\left(\frac{\pi}{2}\frac{t/T+s}{1+s}\right)$ is the cosine noise schedule (Nichol & Dhariwal, 2021), $\mathbf{I} \in \mathbb{R}^{C \times C}$ is the identity matrix, and $\mathbf{1} \in \mathbb{R}^C$ is the one-valued vector. As $t \to T$, the probability for real categorical attributes to be corrupted into random samples from the prior distribution approaches 1.

In addition, we propose two modifications to the diffusion process that better preserve the layerwise patterns of DAGs. To handle the potential uneven graph sparsity with respect to layer depth $l$, we set the prior probability of a directed edge $(u, v)$ for $u \in \mathcal{V}^{(\leq l)}$ and $v \in \mathcal{V}^{(l+1)}$ to be $\frac{\min(|\mathcal{V}^{(\leq l)}|, d_{\mathrm{in}})}{|\mathcal{V}^{(\leq l)}|}$, where $d_{\mathrm{in}}$ is the average node in-degree of the training data. As all nodes in $\mathcal{V} \setminus \mathcal{V}^{(1)}$ have at least one predecessor, we enforce this property in the sampling for graph structure corruption and generation.

To generate $\mathbf{X}^{(l+1)}$, the node attribute prediction module first samples $\mathbf{X}^{(l+1,T_X)} \in \mathbb{R}^{|\mathcal{V}^{(l+1)}| \times C}$ from its prior distribution, where $T_X$ is the maximum number of denoising steps. Then iteratively, it samples $\mathbf{X}^{(l+1,t)}$ with a denoising network $\phi_{\theta_X}\left(G^{(\leq l)}, \mathbf{X}^{(l+1,t+1)}, t+1\right)$ for $t = T_X - 1, T_X - 2, \cdots, 0$. Similarly, the edge prediction module iteratively samples $\mathbf{A}^{(l+1,t)}$ with a denoising network $\phi_{\theta_E}\left(G^{(\leq l)}, \mathbf{X}^{(l+1)}, \mathbf{A}^{(l+1,t+1)}, t+1\right)$.

## B  BASELINE EXTENSIONS

**General extension for DAG generation.** To extend generative models of undirected graphs for DAG generation, we constrain the structure generation to the lower-triangular part of adjacency matrices with a topological ordering.

**GraphRNN.** GraphRNN originally employs rows of an adjacency matrix for both the encoder input and decoder output. Following the practice of (Zhang et al., 2019), we augment them with one-hot encodings of categorical attributes for encoding and predicting the node attributes.

**GraphPNAS.** We use two separate encoders for node attribute prediction and structure prediction as in LayerDAG. Repeatedly, the model first predicts the categorical attributes of a new set of nodes given the partially generated DAG, and then it predicts the incident edges of the new nodes given the partial DAG and the new node attributes. We model the termination of DAG generation as an extra node attribute, which enables the model to generate DAGs with an arbitrary number of nodes different from multiples of the pre-specified size.

We extend the official implementation for D-VAE and GRAN, both use MIT license.

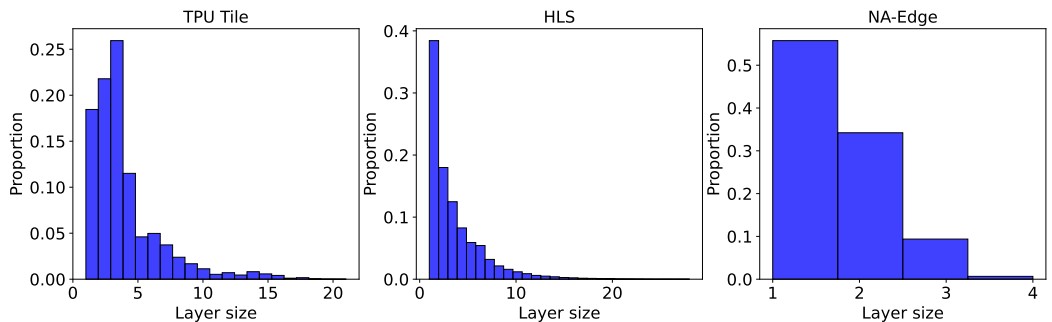

Figure 3: Layer size distribution in the real-world datasets.

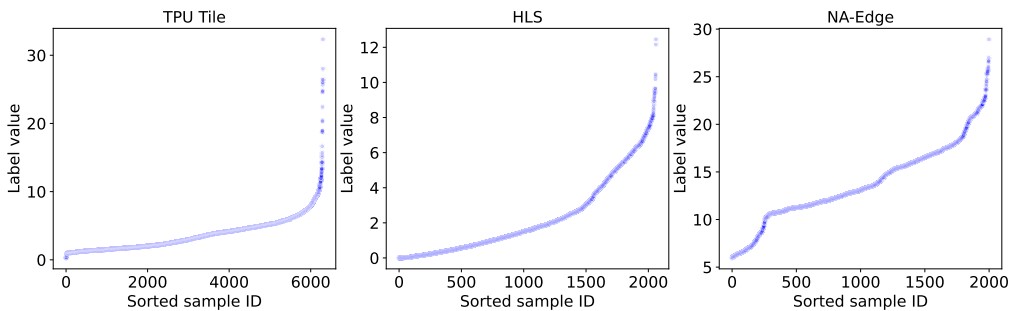

Figure 4: Scatter plot of the sorted labels. The label distributions in the real-world datasets are long-tailed.

## C A SYNTHETIC DATASET OF LATENT PREFERENTIAL DAGS (LP)

A latent preferential DAG is constructed as follows.

1. Randomly sample the number of layers $L \sim \mathcal{U}\{2, 5\}$.

2. Create the first layer of $|\mathcal{V}^{(1)}| \sim \mathcal{U}\{1, 5\}$ nodes. Each node is associated with a random binary attribute $0$ or $1$.

3. For $l = 2, \cdots, L$:

   (a) Create a new layer of $|\mathcal{V}^{(l)}| \sim \mathcal{U}\{1, 5\}$ nodes. Each node is associated with a random binary attribute $0$ or $1$.

   (b) For each new node $v$:

      i. An edge $(u, v)$ is created with a probability $\propto \frac{1}{d_v - d_u}$, where $d_v = l$ is the depth of node $v$ and $d_v > d_u$.

      ii. Let $n_v^{(i)}$ be the number of predecessors of node $v$ with attribute $i$ for $i \in \{0, 1\}$. We enforce that $\frac{\lfloor |n_v^{(0)} - n_v^{(1)}|/2 \rfloor}{(n_v^{(0)} + n_v^{(1)})/2} \leq \rho$, where $\lfloor \cdot \rfloor$ is the floor function.

      iii. $n_v = n_v^{(0)} + n_v^{(1)} \sim \mathcal{U}\{1, 4\}$.

The logical rules for predecessors get increasingly relaxed as $\rho$ goes from $0$ to $1$.

## D VARIANT OF THE LP DATASET FOR ASSESSING THE VALIDITY OF GENERATED NODE ATTRIBUTES

We extend the original synthetic dataset with $\rho = 0$ (full balance requirement). Specifically,

- Each node now has three binary attributes (previously one).

Table 5: Evaluation results on the extended synthetic dataset for $\rho = 0$ (full balance requirement). Best results are in **bold**.

| Model | Validity ↑ | | | $W_1$ / MMD ↓ | | |
|---|---|---|---|---|---|---|
| | balance | attribute | full | $L$ | $|\mathcal{V}^{(l)}|$ | attribute |
| D-VAE | 0.536 | 0.057 | 0.039 | $5.1e^{-1}$ | $1.6e^{-1}$ | $2.5e^{-3}$ |
| GraphRNN | 0.547 | 0.172 | 0.094 | $5.8e^{-1}$ | $1.2e^{-1}$ | $8.5e^{-4}$ |
| LayerDAG | **0.578** | **0.274** | **0.195** | **$1.8e^{-1}$** | **$2.5e^{-3}$** | **$7.8e^{-4}$** |

- For nodes in the first layer, all three binary attributes are randomly sampled from prior Bernoulli distributions.

- The first attribute is still used for determining edge connections based on balance.

- The second and third attributes of an intermediary node are assigned the most and least common corresponding attribute values among its predecessors, respectively. This design is inspired by real-world scenarios, such as tensor dimension matching in computational graphs. In cases of ties, attribute values are assigned randomly.

In addition to the previously adopted metrics, we also report balance-only validity and feature-only validity as key components of the full validity metric, as well as MMD for feature distributions. Table 5 compares LayerDAG with the two most competitive baselines concluded from the other experiments, with results averaged over three random seeds. Overall, LayerDAG achieves the best performance across all metrics. This demonstrates LayerDAG's superior capability in generating valid attributes and learning the attribute distribution.

## E  ADDITIONAL DETAILS ON REAL-WORLD DATASETS AND ADAPTATION OF THEM

**TPU Tile.** In the original dataset, each data sample includes a computational graph, a compilation configuration, and the execution time of the graph on TPU when compiled with that configuration. A single graph may appear in multiple data samples and have multiple associated compilation configurations. We simplify the dataset by averaging the runtime across all compilation configurations for each graph. As the labels of the test set were not released, we re-perform a split of the labeled samples.

**HLS.** We randomly choose 20% of the original graphs.

For all three real-world datasets, we perform a 80/10/10 random split of the dataset. All three datasets exhibit long-tailed layer size distributions (Fig. 3) and label distributions (Fig. 4).

TPU Tile was originally part of the TpuGraphs dataset, which uses Apache-2.0 license. The HLS dataset is not released with a license. NA-Edge is released with an MIT license as part of the nn-Meter project.

## F  EXPERIMENT DETAILS

### F.1  MODEL DEVELOPMENT

For the synthetic LP dataset, we tune the hyperparameters based on validity. For conditional generation, we tune the hyperparameters based on Pearson correlation coefficient. For each experiment, an early stop is performed based on validation accuracies (for layer size prediction) or negative log likelihoods (for diffusion).

### F.2  ML-BASED EVALUATION

We perform ML-based evaluation by implementing a standardized AutoML pipeline. Based on empirical studies, the best BiMPNN model is selected based on validation Pearson correlation coefficient, and the best Kaggle model is selected based on validation mean absolute error.

### F.3 EXPERIMENTS COMPUTE RESOURCES

We have access to an Azure virtual machine equipped with 2 NVIDIA A100 PCIe GPUs, each with 80 GB of memory, 48 non-multithreaded AMD EPYC Milan processor cores, and 440 GiB of system memory.

### F.4 IMPLEMENTATION

Our implementation is based on PyTorch 1.12.0 (Paszke et al., 2019) and DGL 1.1.0 (Wang et al., 2019).

