# OpenReview forum: "LayerDAG: A Layerwise Autoregressive Diffusion Model for Directed Acyclic Graph Generation"
_ICLR.cc/2025/Conference — ICLR 2025 Spotlight_

### Official Review · Reviewer_qSxe · 2024-10-26

**Soundness:** 4
**Presentation:** 4
**Contribution:** 3
**Rating:** 8
**Confidence:** 3

**Summary:**

This submission proposes a generative model, based on autoregression and dffusion, for Direct Acyclic Graph generation (DAG). The problem under study is relevant for the general scientific community with broad applicability in many crucial systems.

The main idea behind the method presented is to transform the topological order induced by the DAG into a series of bipartite graphs that capture sequential dependencies of the tasks represented within the DAG. This idea enables the authors to sequentially generate sets of nodes as if they were a layer of the DAG, overcoming issues with partial ordering of nodes within the topological order of the graph. This is where the autoregressive component of the algorithm comes in, as the generation of a new set of nodes is dependent on the previous layers generated by the model.

The authors propose evaluation on a new synthetic dataset capable of capturing logical dependencies between nodes of different layers with multiple levels of hard logical constraints. This evaluation method attempts to capture the generative model's ability to impose logical dependencies between nodes. The proposed model is further evaluated on the ability to generate DAGs that mimick the properties of DAG property prediction benchmark (supported by an ML-surrogate approach, following related work in the area) and generalization of generated label values. The proposed model shows better performance than related work using the chosen evaluation metrics.

**Strengths:**

- The method is simple, easy to understand and solidly rooted on a wealth of methodology on autoregressive diffusion models.
- The simplicity of the method does not detract the ingenuity of transforming the topological partial ordering into a sequence of bipartite graphs, leveraging that for generation.
- The authors are not over-reliant on mathematical formulation to describe their method (often a problem with diffusion based contributions), striking a balance between textual/intuitive explanations with easy to follow maths.
- Contribution of a synthetic dataset is extremely positive, as it helps testing for a wide range of possibilities that are often not considered with benchmarking datasets (that, in turn, are often not predictive of performance in real-world problems).

**Weaknesses:**

- Although the authors claim a significant increase in the number of nodes generated per DAG (from 24 to 400), this is still far from truly complex DAGs that can number in the thousands.

- There is a lack of critical analysis on the experimental results and the variations in performance of the proposed method. For example, the difference in LayerDAG with a single denoising step to the full algorithm for some benchmarks are negligible, it would be good to have some insight on why and in which conditions is it worth performing multiple denoising steps (and the impact in training time).

**Questions:**

Overall I do not have many questions as I believe this is a high quality submission, however:

- As mentioned in the weaknesses section, I think the paper would be improved by a critical analysis of the experimental results. I understand that space limitations can prevent this, but a high quality discussion of results is more valuable than quantity of benchmarks. For example, how significant is a difference in Pearson correlation of **at most** 0.2 in the generalization benchmark? Is the model able to generalize at all, even if the coefficient is slightly higher than the baselines?

- What is the impact of a BiMPNN instead of considering message passing solely along the direction of the edges?

- If the complexity of generation increases per layer, have you considered using an exponential denoising schedule?

---

> ### Author Response · Authors · 2024-11-24
>
> Thank you for recognizing our contributions and providing insightful feedback!
>
> **Q1**: What is the impact of a BiMPNN instead of considering message passing solely along the direction of the edges?
>
> **Response**: The **effectiveness of BiMPNNs for predicting properties of directed graphs in computing systems** has been established through empirical studies [1]. In system graphs, information flows exhibit intricate patterns where both forward and backward propagation of features can be valuable. Our experimental results support this architectural choice, showing that BiMPNN **either outperforms or matches alternatives for DAG generation tasks**.
>
> [1] Wang et al. A Benchmark on Directed Graph Representation Learning in Hardware Designs.
>
> **Q2**: I think the paper would be improved by a critical analysis of the experimental results, and a high quality discussion of results is more valuable than quantity of benchmarks. For example, how significant is a difference in Pearson correlation of at most 0.2 in the generalization benchmark? Is the model able to generalize at all, even if the coefficient is slightly higher than the baselines?
>
> **Response**: Thank you for this insightful observation. You raise a valid point about the Pearson correlation difference of 0.2 in the generalization benchmark. While this represents a substantial improvement over baselines, we acknowledge that the absolute performance still indicates limited generalization capability. We will revise our manuscript to expand the current discussion. We appreciate your guidance in helping us improve the depth and rigor of our analysis.
>
> **Q3**: If the complexity of generation increases per layer, have you considered using an exponential denoising schedule?
>
> **Response**: We appreciate your insightful suggestion. We agree that exploring alternative diffusion schedules like the exponential denoising schedule is a promising direction for future investigation, potentially allowing better alignment with the complexity scaling.

---

### Official Review · Reviewer_vg1z · 2024-10-27

**Soundness:** 3
**Presentation:** 3
**Contribution:** 3
**Rating:** 8
**Confidence:** 2

**Summary:**

This paper proposes an autoregressive diffusion model to generate directed acyclic graph (DAG). This work observes that the DAG structure can be decoupled by a sequence of tokens, where each token corresponds to a bipartite graph. It is capable of generating larger DAGs (up to ~400 nodes). To validate the model, the paper compares the validity and statistical properties of the synthetic DAGs generated by the model with the baselines: computational graphs on TPU, flow graphs in FPGA, and Neural architectures on edge devices. They use the synthetic DAGs to train a ML-based surrogate model to measure the performance of systems (inference time, resource usage, etc.). And results show that the trained surrogate model outperformed ones that are trained on the baseline generative model.

**Strengths:**

* Make a novel observation that DAG can be viewed as a sequence of bipartite graphs, which preserve the partial ordering and are autoregressive-friendly.
* Explore and discuss different design choices for DAG generative models, including separating three modules, positional encoding, etc.
* LayerDAG is permutation invariant, achieving good generalization.
* Different experiments to evaluate the proposed model quality on different tasks

**Weaknesses:**

* In the beginning of the introduction (line 51), the author mentioned that the one of the challenge in DAG is the training LLM workload execution DAG involves trillions of operations. But this work is capable of generating nodes up to 400 nodes, which seems not to be able to solve the challenge mentioned above.
* In the beginning of the introduction (line 51), the author mentioned that one of the challenges in DAG is the training LLM workload execution DAG involves trillions of operations. But this work is capable of generating nodes up to 400 nodes, which seems not to be able to solve the challenge mentioned above. Or do the authors suggest that using DAG with 400 nodes could capture the characteristics of a trillion of nodes? How do you get the logical constraints from the real-world large DAG for the synthetic DAG generation?
* In D-VAE, they have additional experiments to evaluate the model quality, like reconstruction accuracy. Would like to see the comparison on those experiments.

Minor issue
* The symbol for DAG are not consistent (line 143 vs. line 207)

**Questions:**

Questions
* What is the maximum number of layers of DAG your model could generate?
* Is LayerDAG able to capture more complex logical constraints other than the number of predecessors?
* Do you have explanations for the better performance of baseline models in Table 3, specifically HLS dataset?

---

> ### Author Response · Authors · 2024-11-24
>
> Thank you for reviewing our manuscript and recognizing our contributions!
>
> **Q1**: What is the maximum number of layers for DAGs generated by your model?
>
> **Response**: LayerDAG can generate an unlimited number of layers through enforcing non-zero nodes in subsequent layers. However, **the practical utility may diminish when deviating significantly from real-world graphs**. In our experiments with the evaluated datasets, we observed that the real graphs contain approximately 100 layers at maximum (as detailed in Table 2). Our empirical studies demonstrate that our model achieves competitive performance in capturing the layer count distribution when compared to the baselines (Table 3).
>
> **Q2**: Is LayerDAG able to capture more complex logical constraints other than the number of predecessors?
>
> **Response**: LayerDAG demonstrates strong capability in capturing complex logical constraints beyond predecessor count through our comprehensive empirical evaluation. Our analysis in Section 5.2 shows how the model **effectively handles the intricate interplay between structural dependencies and node attributes like operator types, affecting system performance metrics such as runtime and resource utilization**. We provide further evidence in Appendix D, where controlled experiments with **synthetic datasets were specifically designed to mimic constraints like tensor dimension matching** validate LayerDAG's effectiveness beyond simple predecessor counting. These results from both real-world computational platforms and carefully constructed synthetic scenarios consistently demonstrate that LayerDAG successfully models sophisticated logical relationships in computational graphs.
>
> **Q3**: Do you have explanations for the better performance of baseline models in Table 3, specifically HLS dataset?
>
> Response: While our method achieves the **best overall performance on the HLS dataset**, we acknowledge that some baselines show stronger or comparable results on specific metrics. This behavior can be attributed to the **highly skewed and imbalanced layer size distribution in HLS**, as illustrated in Figure 3. This characteristic likely affects LayerDAG's ability to capture the layer count distribution effectively. However, it's important to note that among all baselines, **only D-VAE outperforms our approach on this specific aspect, while our method maintains superior or comparable performance across other metrics**.

---

> > ### Comment · Reviewer_vg1z · 2024-11-25
> >
> > Thank you for answering the questions and addressing my concerns. I do not have further questions.

---

> > > ### Author Response · Authors · 2024-11-25
> > >
> > > Thank you for reading our responses. We are glad that we've addressed your concerns, and we really appreciate your efforts in reviewing our manuscript!

---

### Official Review · Reviewer_72KY · 2024-10-28

**Soundness:** 3
**Presentation:** 4
**Contribution:** 3
**Rating:** 8
**Confidence:** 4

**Summary:**

The paper presents a diffusion model LayerDAG for directed acyclic graphs (DAGs) based on a novel representation of DAGs as a sequence of bipartite graphs, each bipartite graph divided across nodes in the next layer of the DAG versus nodes in all previous layers of the DAG. This representation allows for layer-wise conditional generation of the DAG, which allows the model to better-learn logical rules. The experiments show that LayerDAG exceeds in learning hard logical rules, learning soft rules, and in generalization.

**Strengths:**

The paper's contributions stem from a novel formulation of a DAG as a sequence of layers. This formulation is original, non-trivial, and concise, and directly motivates the LayerDAG algorithm. The paper itself is quite clear and well-written. The experiments are comprehensive and clearly demonstrate the superior performance of LayerDAG over other autoregressive DAG models.

**Weaknesses:**

A minor weakness in the paper is that there is not enough discussion about the definition of LayerDAG across layers versus the usual definition of diffusion models across denoising steps. I elaborate more in the "Questions" section.

**Questions:**

Pg 6, proof of Proposition 3.1: what is does it mean to be "permutation equivariant"? Is this meant to read "...invariant"?

There is not enough discussion about how exactly the denoising steps interact with the successive layer-wise generation. It is not clear to me how the number of denoising steps relates to the number of layers. It is also jarring to read in S2 that in general diffusion generation is conditioned on previous steps, whereas in LayerDAG it is conditioned on previous layers. For example, the "__Implementation__" paragraph in page 5 reads "Let $\textbf{X}^{l+1,t}$ be the node attributes sampled for the $(l+1)^{\text{th}}$ layer at the step $t$ in the reverse node attribute diffusion process." Is step $t$ nested under layer $l+1$, i.e. all steps $1,\ldots,T$ occur in each layer, or do steps stretch over all layers? Similarly, S3.4 should be better-explained.

---

> ### Author Response · Authors · 2024-11-24
>
> We sincerely thank the reviewer for the thorough evaluation and valuable feedback!
>
> **Q1**: In Pg 6, proof of Proposition 3.1: what does it mean to be “permutation equivariant”? Is this meant to read “...invariant”?
>
> **Response**: Thank you for this important clarification question. The term "permutation equivariant" is indeed the correct terminology here, distinct from "permutation invariant." To achieve permutation invariance in the learned distribution of graphs, we require a graph encoder that computes and updates node representations in a permutation equivariant manner [1, 2]. This means that **if we permute the ordering of nodes in the input graph structure and their corresponding features, the computed node representations must undergo the exact same permutation**. This fundamental property ensures ordering-agnostic behavior, and the permutation equivariance of the encoder serves as a necessary prerequisite for achieving permutation invariance in the overall learned distribution. We can observe this property in many message passing neural networks. Consider the basic matrix operation $AX$, where $A$ denotes the adjacency matrix and $X$ represents the input node features – this operation inherently preserves permutation equivariance.
>
> [1] Srinivasan & Ribeiro. On the Equivalence between Positional Node Embeddings and Structural Graph Representations.
>
> [2] Want et al. Equivariant and Stable Positional Encoding for More Powerful Graph Neural Networks.
>
> **Q2**: How does the number of denoising steps relate to the number of layers? In particular, for “**Implementation**” paragraph in page 5, is step $t$ nested under layer $l+1$, i.e., all steps $1, \cdots, T$ occur in each layer, or do steps stretch over all layers?
>
> **Response**: In our framework, $T$ denotes **the number of diffusion steps that occur within each layer, i.e., all steps $1, \cdots, T$ occur in each layer**. This notation choice effectively captures the interplay of layerwise autoregressive generation and intra-layer diffusion. We appreciate your suggestion and will refine the presentation for better clarity in the next version of our manuscript.

---

> > ### Comment · Reviewer_72KY · 2024-11-25
> >
> > Thanks for the responses. No further questions.

---

> > > ### Author Response · Authors · 2024-11-25
> > >
> > > Thank you for reading our responses and we are glad that we've addressed your questions!

---

### Official Review · Reviewer_7SrQ · 2024-10-29

**Soundness:** 3
**Presentation:** 3
**Contribution:** 3
**Rating:** 6
**Confidence:** 3

**Summary:**

The paper makes a compelling contribution by proposing a novel framework for DAG generation that addresses key challenges in modeling both logical and directional dependencies. The combination of autoregressive generation and diffusion models offers strong expressiveness, and the extensive experiments validate its practical utility. However, addressing the computational complexity and providing more interpretability would strengthen the work. Expanding the comparison with simpler models and exploring alternative encoding methods would also improve the empirical analysis. Nonetheless, the novelty and relevance of LayerDAG justify its acceptance.

**Strengths:**

Innovative Approach: The combination of autoregressive and diffusion models provides a novel framework for generating complex DAGs with strong dependencies.

Layerwise Decomposition: The use of bipartite graphs for sequential layer generation improves scalability and reduces computational complexity.

Conditional Generation Capability: LayerDAG supports generating DAGs based on specific performance metrics, making it valuable for system benchmarking applications.

**Weaknesses:**

Computational Overhead: While the layerwise approach is efficient, the combination of autoregressive and diffusion methods can be computationally intensive for very large graphs.

Limited Exploration of Alternative Encodings: The paper mainly focuses on sinusoidal encodings, without exploring other potential positional encodings.

**Questions:**

NA

---

> ### Author Response · Authors · 2024-11-24
>
> Thank you for your positive and detailed feedback!
>
> **Q1**: Computational Overhead: While the layerwise approach is efficient, the combination of autoregressive and diffusion methods can be computationally intensive for very large graphs.
>
> **Response**: Thank you for this thoughtful observation. We directly tackle this challenge through our **non-uniform denoising schedule**, detailed in Section 3.4, which strategically **allocates more diffusion steps/computation budget to deeper layers** where the generation requires dealing with more complex dependencies. Our empirical studies in Section 5.4 demonstrate its effectiveness in **balancing computational efficiency and generation quality**. We agree that extending these experiments to even larger directed acyclic graphs (DAGs) represents a valuable direction for future research, and our framework provides a foundation for such investigations.
>
> **Q2**: Limited Exploration of Alternative Encodings: The paper mainly focuses on sinusoidal encodings, without exploring other potential positional encodings.
>
> **Response**: We appreciate the reviewer’s observation regarding positional encodings. Our choice of sinusoidal encodings was motivated by their proven expressiveness and generalization capabilities across various scenarios. Nevertheless, we acknowledge that exploring alternative encoding schemes could yield valuable insights in future studies.

---

### Official Review · Reviewer_MRjN · 2024-11-04

**Soundness:** 3
**Presentation:** 3
**Contribution:** 2
**Rating:** 6
**Confidence:** 3

**Summary:**

This paper presents an autoregressive model, LayerDAG, for generating directed acyclic graphs (DAGs). A key contribution of the paper is framing the problem as a layered graph, leveraging the fact that a topological ordering of a DAG naturally induces a layered structure, where each pair of consecutive layers forms a directed bipartite graph. Building on this layered representation, LayerDAG employs autoregressive generation to model directional dependencies between layers, while diffusion models capture logical dependencies within each bipartite graph. Experimental results demonstrate that LayerDAG outperforms several existing graph generative models.

**Strengths:**

1.	Generating DAGs is more challenging than generating general graphs due to their inherent dependencies and constraints. The layered structure simplifies the generation process by breaking down the DAG into a sequence of directed bipartite graphs, making dependencies easier to handle.
2.	LayerDAG's combination of autoregressive and diffusion models makes it a more robust approach to handling these complexities.
3.	LayerDAG offers a scalable solution for generating large, structured DAGs.
4.	The paper demonstrates the improved performance of LayerDAG over other methods through experiments with various synthetic DAGs.

**Weaknesses:**

1.	The authors state that "LayerDAG is based on a novel perspective of DAGs: as illustrated in Fig. 1(b), the partial order of nodes dictated by the DAG structure can be decoupled as a sequence of tokens, each corresponding to a bipartite graph." However, this approach has been widely known for decades. For example, Tarjan’s topological ordering algorithm for a DAG uses a Depth-First Search (DFS) traversal to produce a linear ordering of vertices, ensuring that for every edge, the source vertex appears before the target vertex in the ordering. Surprisingly, the paper does not reference Tarjan’s method or other relevant work in this area.
Citation: Tarjan, R. E. (1972). Depth-first search and linear graph algorithms. SIAM Journal on Computing, 1(2), 146–160. doi:10.1137/0201010
2.	The directional and ordered dependencies within a DAG structure are essential, meaning that order should be preserved. Thus, the motivation for seeking permutation invariance in this context is unclear. Furthermore, Proposition 3.1 lacks clarity, and a formal proof would help understand the proposition.
3.	In Section 5.2, the experiments explore several real-world DAGs. However, the presented results indicate whether the generated DAGs can substitute training DAGs, which seems insufficient. For instance, the generated graphs have different layer counts than the actual DAGs. A more pertinent question would be whether the generated DAGs are suitable for downstream tasks. Although validating hardware DAGs might be challenging, it should be feasible to assess the relevance of other types of DAGs for specific applications (e.g., with the TPU Tile dataset).

**Questions:**

1.	Given that the topological ordering of a DAG is not unique, how does the layered representation remain consistent for a given DAG? Was any consideration given to the ordering of nodes within each layer?
2.	What is the rationale for seeking permutation invariance when generating DAGs?
3.	Is it possible to validate the generated DAGs for the TPU Tile datasets experimentally, without relying on a surrogate model?

---

> ### Author Response · Authors · 2024-11-20
>
> Thank you for carefully reviewing our manuscript and providing constructive feedback. Below we respond to your questions and concerns.
>
> **Q1**: Is the proposed sequential view of directed acyclic graphs (DAGs) merely topological ordering? If so, how do you handle the non-uniqueness of topological orderings for autoregressive generation? How is this related to permutation invariance?
>
> **Response**: Our proposed sequential view of directed acyclic graphs (DAGs) extends beyond classical topological ordering. While a topological ordering arranges vertices linearly such that a vertex u precedes vertex v if there is an edge (u,v), its non-uniqueness poses a challenge for autoregressive DAG generation. Specifically, models like D-VAE [1] rely on randomly selected topological orderings during training, violating permutation invariance: the probability of generating the same DAG varies for different topological orderings. This compromises generalizability and necessitates costly strategies like sampling diverse orderings.
>
> In contrast, our approach guarantees ordering uniqueness by generalizing from vertex ordering to vertex **set ordering**. Instead of sequentially generating individual vertices in a specific ordering, we construct **layers** — vertex sets whose members are characterized by the absence of unvisited predecessors. This results in a unique, sequential ordering of vertex sets (or layers), ensuring consistency across representations. The proposed LayerDAG framework leverages this unique layer structure to perform autoregressive generation at the level of vertex or edge sets, rather than individual vertices or edges.
>
> By adopting this **unique** layer-based construction, our model achieves **permutation invariance**: the generation probability of a DAG remains unchanged regardless of the topological ordering of vertices. This eliminates the need to sample diverse orderings during training, potentially improving both training efficiency and model quality. Further details on this approach can be found in Sections 3.1 and 3.2 of the manuscript.
>
> **Q2**: The current generation evaluations are based on statistics and discriminative/surrogate models. Is it possible to directly validate and assess the generated DAGs on the target computational platforms like TPU?
>
> **Response**: Thank you for your insightful feedback! We choose to establish the testbed based on surrogate models as they are more accessible to the machine learning (ML) community in general and allow the ML community to easily further the research efforts. For example, real-world validation of the generated graphs for TPU Tile requires both access to TPUs and expertise about deep learning compilers. In fact, the system community also heavily employs ML-based surrogate models today, and we list diverse examples in the Section 5.2.
>
> We agree that ultimately this line of research efforts is most useful if the generated synthetic DAGs can effectively replace real DAGs on the end computational platforms. We are currently working with experts in system and compute architecture for performing more domain-specific evaluations. We intend to submit the extension work directly to system/computer architecture venues where the audience can more effectively assess the work for domain purposes.
>
> **Q3**: Why didn’t you cite the classical papers on topological ordering like Tarjan, R. E. (1972)?
>
> **Response**: We have revised our manuscript to cite two classical papers on topological ordering including Tarjan, R. E. (1972) at the beginning of Section 3.1. We appreciate your guidance in helping us properly attribute the credit.
>
> [1] Zhang, M et al. D-VAE: A Variational Autoencoder for Directed Acyclic Graphs. NeurIPS 2019.

---

> > ### Comment · Reviewer_MRjN · 2024-11-24
> >
> > I want to thank the authors for answering my questions. I agree with them that direct validations without surrogate models could be useful to showcase the method's effectiveness.  Please consider adding proof or formal validation for Proposition 3.1. Thank you.

---

> > > ### Author Response · Authors · 2024-11-24
> > >
> > > Thank you so much for reading our response and guiding us to improve our work! We will incorporate your suggested changes for the final version.

---

### Author Response · Authors · 2024-11-25

We sincerely appreciate the insightful and positive feedback from all reviewers and the great coordination efforts of the area chair. Specifically,

- All reviewers recognized the contribution and novelty in decomposing a directed acyclic graph (DAG) into a unique ordered sequence of node and edge subsets, effectively generalizing classical topological orderings and addressing the issues arising from their non-uniqueness in existing autoregressive DAG generative models.
- The novel sequential view above naturally leads to the design of a novel autoregressive diffusion model of DAGs (Reviewers MRjN and 7SrQ).
- Empirical studies demonstrate the effectiveness of the proposed approach (Reviewer 72KY) for synthetic graph generation (Reviewer MRjN), conditional generation with real-world DAGs for system benchmarking (Reviewers 7SrQ), achieving scalability (Reviewers MRjN and 7SrQ), and generalizability (Reviewers vg1z).
- Reviewer qSxe recognized the contribution of a new synthetic dataset for advancing future research.
- All reviewers find the paper well-written and the presentation easy to follow, “striking a balance between textual/intuitive explanations with easy to follow maths” (Reviewer qSxe).

We’ve responded to individual reviewers for the specific questions. Three reviewers have already responded and acknowledged that we’ve addressed their questions (MRjN, 72KY, and vg1z).

---

### Meta-Review · Area_Chair_rQ5f · 2024-12-17

**Metareview:**

This paper introduces LayerDAG, a novel approach for generating directed acyclic graphs (DAGs) that combines autoregressive generation with diffusion models. LayerDAG leverages the layered structure of DAGs, induced by their topological ordering, to model both directional and logical dependencies effectively. The authors demonstrate the superior performance of LayerDAG compared to existing graph generative models, particularly in capturing complex dependencies and generating large DAGs.

Reviewers praise the innovative combination of autoregressive and diffusion models in LayerDAG, recognizing its potential to improve the expressiveness and scalability of DAG generation. The empirical results convincingly demonstrate the advantages of LayerDAG over existing methods, particularly in capturing complex dependencies and generating large, structured DAGs. The layered representation is seen as a key enabler for achieving these improvements.

However, one weakness is also identified:

- Computational Overhead: While the layerwise approach offers efficiency gains, the combination of autoregressive and diffusion models can lead to increased computational costs, particularly for very large graphs.

Recommendation:

Despite the computational considerations, the reviewers agree that LayerDAG presents a significant advancement in DAG generation. So I would suggest to accept it as poster.

Program Chair Modification: The area chair initially accepted, then rejected the paper later. However, this rejection was not confirmed by the Senior AC. ICLR requires that rejecting a paper with otherwise unanimous reviews requires Senior AC confirmation. The paper is thus accepted.

**Additional Comments On Reviewer Discussion:**

The discussion was smooth and nice.

---

### Decision · Program_Chairs · 2025-01-22

Accept (Spotlight)